# The Perceptions of and Attitudes Toward Obesity in Bulgarian Adults with a BMI ≥ 25.0 kg/m^2^—An Exploratory Study

**DOI:** 10.3390/nu17030373

**Published:** 2025-01-21

**Authors:** Mihail A. Boyanov, Margarita B. Grigorova, Anna T. Karteva-Stoycheva, Todorka K. Atanasova, Maria G. Nikolova

**Affiliations:** 1Clinic of Endocrinology and Metabolism, University Hospital Alexandrovska, St. Georgi Sofiysky Str. 1, 1431 Sofia, Bulgaria; mihailboyanov@yahoo.com; 2Department Internal Medicine, Faculty of Medicine, Medical University of Sofia, St. Georgi Sofiysky Str. 1, 1431 Sofia, Bulgaria; 3Noema Company, 1000 Sofia, Bulgaria; margarita.noema@gmail.com; 4CMR Department, Novo Nordisk EAD Bulgaria, Zlaten Rog Str. 20, 1407 Sofia, Bulgaria; nkva@novonordisk.com; 5Obesity Department, Novo Nordisk EAD Bulgaria, Zlaten Rog Str. 20, 1407 Sofia, Bulgaria; 6Department of Epidemiology and Hygiene, Faculty of Medicine, Medical University of Sofia, St. Georgi Sofiysky Str. 1, 1431 Sofia, Bulgaria

**Keywords:** obesity, interview, perceptions, attitudes, discrepancies

## Abstract

Background: Obesity affects 33.2% of the adult population in Bulgaria, and there is a scarcity of information about affected individuals’ attitudes toward their weight situation. The aim of this study was to explore the perceptions of obesity in affected adults. Methods: The present study involved a questionnaire-based survey that utilized home-based and tablet-assisted face-to-face interviews. Interviewees comprised individuals aged 25–64 y/o with a BMI > 25.0 kg/m^2^. Results: Overall, 704 respondents participated (344 overweight; 360 obese). Over 50% of participants reported attempts to reduce their weight, with only 6% of participants in the overweight group and 16% in the obese group perceiving their condition as worrisome. One-third of the obese participants considered their state temporary. The main cause for alarm in overweight/obese participants was a worsening overall physical condition in males and an increase in clothes size in females. The need for an urgent reduction in body weight was noted by 12% of the overweight respondents and 40% of the obese respondents. The reasons for being overweight were collated as a lack of physical activity (noted by 52% of participants), sedentary lifestyle (51%), stress/depression (41%), excessive consumption of carbohydrates (34%), general overconsumption of food (33%), and poor quality of food products (28%). Of note, 56% of the respondents had first consulted a medical professional about being overweight. Most respondents selected dieting without physical activity for weight reduction, with 48% stating that they would try drugs approved for weight reduction. Conclusions: Many overweight/obese individuals have an unrealistic self-perception and very low motivation to take active measures. These discrepancies offer great opportunities for better public education and structured, active strategies.

## 1. Introduction

Obesity is an increasingly prevalent contemporary health epidemic affecting more than 1 billion individuals worldwide, with this number projected to rise in the coming decades [1]. An increasing number of individuals are living in better environments than those 20–30 years ago, consuming more calories, and leading a more sedentary lifestyle. At the other end of the problem is the food industry, which contributes to the problem by offering increasingly processed foods and exploiting inborn human instincts, such as appetite, satiety, and reward, for profit. The pathological sequelae of obesity include numerous adverse medical conditions and life-debilitating diseases. Obesity is not only one of the main components of metabolic syndrome, but it is also a driving force for associated conditions, such as type 2 diabetes, atherogenic dyslipidemia, and hypertension. It is therefore implicated in the origin of cardiovascular and cerebrovascular diseases, as so often seen in affected individuals [2]. One of the first medical complications of excessive body weight is osteoarthritis, which shows greater levels of treatment resistance in overweight than in normal-weight individuals [3]. Obstructive sleep apnea and obesity-induced hypoventilation syndrome are other potentially life-threatening complications, even in the very young [4]. Accumulating evidence has revealed the crucial role of obesity in the progression of chronic kidney disease and metabolic fat disease [5,6]. Notably, obesity has emerged as a strong risk factor for cancer and negatively affects its course and prognosis [7]. As a result, obesity increases morbidity and mortality and incurs both human and economic costs to all of society [8]. The authors of a recent study conducted in Canada performed a fiscal analysis and showed that small reductions in the prevalence of obesity (such as 1–2%) may result in positive fiscal gains for the government from reduced spending on public benefits, together with increased tax revenue due to better employment status [9]. Similarly, an expansive US electronic medical record database was used to create a model assessing direct treatment costs and the cost of additional obesity-related diagnoses [10]. The results of this analysis confirmed that weight loss in this population may correspond to reduced incident obesity-related complications and therefore lead to substantial cost savings [10].

Typically, doctors consider obesity a common disease that can be treated by lifestyle changes and by simply prescribing drugs [11,12]. However, the situation seems far more complex. First, the affected person should regard his/her condition as a disease, not as a minor health complication or cosmetic problem. Second, lifestyle changes require deeply founded motivation and strong willpower from the patient. Awareness and incentives are needed on both sides—the patient’s and the medical specialist’s. Third, there remains a scarcity of drugs to effectively treat obesity, with prices that are above those of more common medications.

Very little is known about the perceptions of people who are overweight/obese regarding their condition. These perceptions are the result of multiple factors: personal beliefs and character traits, sources of available information, understanding by the individual, family support, financial limits, and many other factors [13,14,15]. Emotional dysregulation may be both a causative factor and a consequence of excessive body weight [16]. Stigmatization, often starting at a very young age, may profoundly affect the self-perception and beliefs of obese individuals [17].

Obesity is a common health condition in Bulgaria. The 2012 National Representative Epidemiologic Survey showed that 33.2% of the adult population were obese (28.3% of females and 38.8% of males) and 37.0% of the entire sample were registered as overweight (32.5% of females and 42.2% of males) [18]. A total of 62% of the participants had an abnormal waist circumference (63.2% of females and 61.6% of males) [19]. Almost no information is available about the affected individuals’ attitudes toward their personal condition, the way in which they gather information about their condition, their personal perceptions about the need for urgent action, and the ways they can proceed. The medical community is in desperate need of more detailed information about the concerns, fears, and hopes of those who are overweight/obese to properly guide management and treatment decisions.

The aim of this study was to explore the perceptions of and attitudes toward obesity in Bulgarian adults who are overweight or obese.

## 2. Materials and Methods

### 2.1. Study Design and Ethical Standards

This study comprised a questionnaire-based survey, primarily conceived as a marketing tool to explore different groups of individuals with obesity. According to local regulations, ethical approval is not required for questionnaire-based surveys. The ethical standards approved by the European Society for Opinion and Marketing Research were strictly followed. All respondents were volunteers and gave their informed consent prior to the initiation of any procedures. Their personal data were anonymized. The marketing research agency carrying out the fieldwork was certified as an operator of personal data by the Bulgarian authorities.

First, a pilot (target) group completed a qualitative survey to better formulate the research questions. The interview lasted 3 h and covered a number of topics: free and leisure time, attitudes toward “good health” and “natural ways to lose weight”, and sources of information related to weight loss. The results were then used to structure the computer-based questionnaire for the large study group.

In the second stage, the actual quantitative home-based interview was completed using computerized tablets. All eligible subjects took part in this in-home, tablet-assisted, and face-to-face interview. Interviews lasted an average of ~25 min. Participants first gave their informed consent for data collection, analysis, and storage, with the in-home interviews being performed between 20 January and 5 February 2022.

### 2.2. Participants

The participants comprised male and female volunteers aged between 25 and 59 years. The following inclusion criteria were used:Aged between 25 and 59 years;BMI ≥ 25.0 kg/m^2^;Household monthly income above the Bulgarian average (BGN ≥ 2400 ≈ EUR ≥ 1200 for the pilot group and BGN ≥ 3000 ≈ EUR ≥ 1500 for the actual survey participants). The respondents were divided into 2 cohorts based on their age: 25–39 y/o and 40–59 y/o. The participants’ body mass index was calculated based on self-reported height and weight during recruitment and following the interview. The income criteria were introduced to select participants able to afford more expensive diets, food supplements, or drugs typically utilized in the management of obesity.

The target (pilot) group consisted of 4 online group discussions, each with 6 participants—2 participants with a BMI between 25 and 27 kg/m^2^ and 4 participants with a BMI > 27.0 kg/m^2^. The home-based interview group (quantitative survey) included individuals aged 25–64 y/o with a male-to-female ratio of roughly 50:50, all with a BMI > 25.0 kg/m^2^. The survey covered the 6 largest cities in Bulgaria (Sofia, Plovdiv, Varna, Burgas, Stara Zagora, and Blagoevgrad) with a cumulative total of 1.3 million inhabitants (34% of the country’s population). Three hundred and sixty thousand inhabitants (28% of the cities’ population) were registered as having a higher-than-average household income. One hundred and seventy thousand individuals in this segment (47% of the cities’ population with a higher income) had a BMI ≥ 25.0 kg/m^2^, based on previous survey results, with 17% being in the BMI stratum of 25.0–29.9 kg/m^2^. The sample of participants was distributed on a random basis throughout the six cities. A randomly selected street within the city, coupled with a randomly selected address, served as a starting point, with the interview area steadily increasing outward. Interviewers systematically visited every address, engaged with a member of their household, and ensured that they answered the relevant sections of the questionnaire at the designated location. From the approximately 4000 addresses visited, a total of 704 effective interviews were collected.

### 2.3. Questionnaire

The patient questionnaire was specifically designed for this study. It included an introductory section with screening questions, followed by 4 sections covering all topics of interest: awareness of the health condition, being overweight and the patient’s journey (how did they end up speaking to a medical professional regarding their condition?), attitudes toward losing weight and the utilization of information channels, and additional demographic data. The questionnaire is available in Appendix A.

### 2.4. Statistical Analysis

The sample size computation was based on additional published data regarding the prevalence of obesity and being overweight in the general Bulgarian population based on data collected in 2012 [8]. The expected representative sample size was calculated to be around 0.4% of the target population (N = 700). This sample size was expected to reach a statistical power of β = 0.8 and significance of *p* ≤ 0.05.

All analyses were performed on an IBM SPSS 19.0 for Windows platform (SPSS Corp., Chicago, IL, USA). Descriptive statistics (means, medians, standard deviations, and quartiles) and frequency analysis were performed. The different shares of patients are presented in percentages. The total sample was subdivided according to BMI (25.0–26.9, 27.0–29.9, and BMI ≥ 30.0 kg/m^2^). Appropriate graphs were constructed. Statistical significance was set as *p* ≤ 0.05.

## 3. Results

### 3.1. Demographic Profile of the Respondents

The survey had a sample size of N = 704, distributed in two equal clusters—individuals with a BMI of 25.0–29.9 and individuals with a BMI ≥ 30 kg/m^2^, randomly selected in each cluster. At least 50% of the overweight participants had made attempts to reduce their weight within the last 2–3 years.

The city of Sofia was represented by 48% of participants, with the second largest city, Plovdiv, represented by 16% and the third largest, Varna, represented by 15%. The remaining three cities constituted 8%, 7%, and 6% of the survey respondents. Table 1 summarizes the sample structure and profile in total and stratified according to BMI.

### 3.2. Overall Health and Concomitant Diseases—Participant Evaluation

The patients’ overall health evaluation was predominantly good or moderate, as shown in Figure 1.

As shown in Figure 1, overall health perception considerably worsened with a BMI ≥ 30. Roughly 37% of all participants reported on average two associated chronic illnesses. Their prevalence in participants with comorbidities (in decreasing order) was as follows: blood pressure (66%), hypercholesterolemia (28%), osteoarthritis (27%), depressive disorders (16%), dysglycemia (14%), heart disease (13%), type 2 diabetes (10%), sleep apnea/snoring (8%), kidney disease (5%), liver disease (2%), and other conditions (12%). Adults 55–64 y/o (N = 154) showed both the highest rates of obesity (60% with BMI > 30 kg/m^2^) and chronic illnesses (68%).

### 3.3. Self-Perception of Being Overweight/Obese and the Urgency to Lose Weight

The attitudes of the respondents toward the problem of being overweight/obese are displayed in Figure 2.

Figure 2 underlines the fact that only one-third of obese individuals considered their state as temporary, with the majority considering their condition permanent. Females and respondents aged 55–64 were more inclined to consider being overweight as permanent and not temporary.

The cause for alarm in overweight/obese participants was a worsening overall physical condition in males; in comparison, females noted an increase in their clothes size as the cause for alarm.

The patients’ views about their need to urgently reduce their body weight is reflected in Figure 3.

### 3.4. Perceived Reasons for Being Overweight/Obese

The reasons for being overweight were rated by the respondents in the following order: lack of physical activity (selected by 52% of participants), sedentary lifestyle (51%), stress/depression (41%), excessive consumption of carbohydrates (34%), general overconsumption of food (33%), poor quality of food products (28%), hormonal changes (24%), and genetic predisposition (21%). Table 2 summarizes the different reported reasons for obesity/being overweight by the subcategories of BMI.

The interaction between obesity and chronic illnesses was perceived in both directions. Twenty-nine percent tended to explain their chronic diseases by linking them to being overweight/obese; however, a large proportion had the belief that obesity is a result of chronic illness.

### 3.5. Preferred Information Sources About Being Overweight/Obese

Of note, 56% of the respondents had first consulted a doctor about being overweight, 27% had consulted a fitness instructor, and 26% had consulted a nutritionist. When asked if the medical specialist (doctor) had paid attention to their weight situation, only 32% of the total sample and 49% of the obese respondents gave a positive answer. This finding shows the low commitment of medical specialists to the issue of being overweight/obese.

Conversely, only 67% of the entire study sample used different sources to obtain information regarding weight loss. Of the respondents, 42% asked friends and relatives in order to become informed about approaches to combat being overweight, whereas 26% and 25% had consulted a doctor or social media, respectively. Moreover, 13% of respondents had consulted a nutritionist and 8% had consulted a fitness instructor. For the obese participants, the doctor was considerably more trusted than social media, though equally used. Younger participants showed a declining trend in consulting a doctor. General practitioners represented the main specialists consulted (in 39%), followed by endocrinologists (29%), nutritionists (14%), and cardiologists (13%). In addition, 10% had consulted another specialist, mainly fitness instructors and dieticians. Once deciding to lose weight, only 22% of participants (32% of those with a BMI ≥ 30 kg/m^2^) had consulted a doctor; in comparison, 55% of the entire sample size proceeded on their own. Table 3 shows a list of different medical specialists that were most trusted for help with weight loss.

Only 16% of respondents reported that they could rely on their families to support their efforts for weight reduction. The closest attitude of friends and family toward their personal weight situation was assessed by the respondents as follows: accepting without interfering (57%), disapproving but not commenting (19%), ready to provide any help (19%), and even making inappropriate jokes (8%). The majority of the families of obese people were willing to interfere, unlike families of people with a lower BMI.

### 3.6. Management of Overweight/Obesity—Knowledge and Attitudes

When asked about their awareness of methods for weight reduction, most participants selected the following responses: diet without physical activity (84%), followed by exercise/physical activity only (80%), OTC drugs and supplements (77%), cosmetic procedures or massage (65%), fitness/gym-based weight loss programs (60%), and surgical interventions (53%). Of note, 48% stated that they would try drugs approved for weight loss, with 31% being aware of injection therapy. Moreover, 61% of the obese individuals were currently trying to lose weight, and 78% of the total sample and 82% of those with a BMI ≥ 30 kg/m^2^ had attempted to lose weight at some point. The main reasons for not trying were listed as feeling good enough (even with excess weight) and lack of will. Figure 4 displays the preferred method for losing weight (if ever tried) as selected by the respondents. On average, two methods had been utilized by those wishing to lose weight.

It was noted that failure to lose the required weight resulted in self-blame, owing to the lack of persistence and lack of will. Regaining the lost weight (kg) over the course of time resulted in many respondents giving up trying and simply resigning themselves to being overweight. Keeping the lost kilos off long-term was considered a much more substantial result than primary weight loss.

The most frequently considered benefits of obesity drugs were safety (73%), efficacy (69%), and positive impact on cardiovascular risk factors (54%). The perceived barriers included fear of weight increase following treatment discontinuation (63%), high drug prices (48%), fear of needles/injections (43%), and the need for long-term use (40%). This behavior resulted in a rather low willingness to buy expensive anti-obesity drugs—overall, only 14% agreed to follow such a course—including 19% of those with a BMI ≥ 30 kg/m^2^.

## 4. Discussion

As a part of our study, we gathered detailed information about common beliefs, knowledge, and attitudes toward being overweight/obese in a representative sample of the urban Bulgarian population. The overall study included a small pilot sub-study (qualitative stage) and the actual quantitative study, which included 704 participants with a BMI ≥ 25.0 kg/m^2^. All participants had at least a middle-to-high income level, and in roughly half of participants, their self-evaluation of their overall health was good, with a minority reporting poor health (<7% for those with a BMI of 25.0–29.9 kg/m^2^ and 16% for those with a BMI ≥ 30 kg/m^2^). Around two-thirds of obese participants (67%) perceived their condition as permanent, with only one-third (33%) seeing urgency in losing weight. Obese respondents expressed a wish to lose a mean of 19.2 kg, a quite difficult goal to achieve. Reported family support for these efforts to lose weight was very low (only 16%), although it was noted that family and friends were the preferred sources for gathering information about possible approaches. Help from doctors was sought in only 26% of cases, followed by nutritionists and endocrinologists as the preferred specialists. Of note, diet and physical activity were perceived by more than 80% of respondents as the main method to combat obesity. However, most respondents chose one single method from the two and very rarely a combined approach. At least 80% of the study sample had made previous attempts to lose weight, mostly with partial and temporary success. The most common perceived barriers were good well-being while being overweight and lack of will. In short, the results of this study show that most individuals who are overweight/obese have an unrealistic self-perception and very low motivation to take any active measures. From the survey responses, help received from family members or medical specialists also seemed to be insufficient. These discrepancies offer great opportunities for better public education and the implementation of structured strategies for fighting the growing prevalence of obesity and being overweight—with easier access to trusted information, specialists, and management opportunities.

The first barrier to obesity management might be the fact that most patients, and a substantial proportion of healthcare practitioners, do not recognize a high BMI as a condition requiring special attention. In the Spanish cohort of the International ACTION-IO observation study, 59% of the affected participants agreed that obesity is a chronic disease [10]. As an additional obstacle, 80% assumed complete responsibility for their own weight management, coupled with a mean delay of 6 years in first discussing this particular problem with a healthcare provider [20]. In the US National ACTION study, 65% of the respondents recognized obesity as a disease; however, only 54% worried about the possible repercussions of excessive weight on their health [21]. Similarly, in our study sample, the majority of participants with a BMI ≥ 30 kg/m^2^ perceived their health status as good or moderate (30% and 54%, respectively) and being overweight as a chronic condition (67%). The authors of recent studies from other populations report similar findings. Among 300 patients with a BMI > 30 kg/m^2^ in the Lazio region, Italy, only 49% correctly identified themselves as being obese [22]. The same percentage (49%) of obese participants in a study in Bahrain perceived themselves as living with obesity [23]. Lower proportions of patients being unaware of their weight situation (30%) were reported in the ACTION-China study [24].

Similar perceptions and attitudes of individuals who are overweight/obese were revealed in the ACTION international and national studies. A very detailed study from Lithuania showed that most participants failed to visually recognize obesity (males more often than females) and had little knowledge about diseases associated with obesity apart from heart disease and diabetes [25]. The findings of this study corroborated our findings that a high proportion of obese individuals do not perceive a problem at all and would not see any urgency in losing weight. Of note, “low metabolism” was perceived as one of the main causes for putting on weight in the Lithuanian population, rated closely after “eating too much” and “exercising too little”. Younger respondents (age < 45 years) were more likely to change their lifestyle. Disappointment with one’s current weight was identified as the driving force for implementing weight loss strategies [25].

The perceived causes of weight gain are also poorly understood, both by patients and healthcare providers. A study examining patients’ perspectives reported genetic (uncontrollable) causes of obesity in 13% of the participants, with the remaining 87% emphasizing the importance of controllable factors (lifestyle habits and unhealthy diet) [15]. Of note, our respondents highlighted the poor quality of food, irregular meals, and a sedentary lifestyle as major causes of obesity.

The personal hopes of losing weight by people living with obesity should also be taken into consideration. Participants with a BMI > 30 kg/m^2^ in our study sample believed that they were overweight by a mean of 20.6 kg and wished to lose 19.2 kg on average, possibly in an urgent manner (40%). In the OBSERVE study, the median preferred percent weight reductions in overweight/obese participants were 23.5% (acknowledged as a “dream”), followed by 16.7% (noted as a “goal”), 14.6% (marked as “happy”), and 10.3% (regarded as still acceptable) [26]. These expectations for losing weight might only be met by recently developed anti-obesity medications and bariatric surgery.

The perceptions and expectations of individuals with obesity were also very well highlighted in the ACTION-IO study [27]. Data from 14,502 obese individuals were summarized and the results showed that only 68% agreed that obesity was a disease. A median of three (mean six) years was needed for the patient to seek medical consultation and the primary reason for this delay was that most (81%) assumed complete responsibility for losing weight. Interestingly, most of the respondents (68%) were inclined to wait for healthcare professionals to address the issue of body weight. In line with our data, most respondents in the ACTION-IO study shared the view that they “ate too much” (62%) and “exercised too little” (73%). Almost two-thirds of the international study population had not tried to lose weight during the last 3 years, which is in contrast with reported attempts to lose weight in more than three-fourths of our study sample [27]. The authors of the ACTION-IO study concluded that several gaps in obesity care still exist together with a misalignment between perceptions and attitudes, a finding corroborated by our data.

The authors of another study examined nutritional knowledge and attitudes and identified self-regulation as a potent predictor of weight problems [14]. Similar to the results reported above, only 28.1% of the participants correctly identified the body mass index cut-off mark for obesity. The participants identified poor eating/self-regulation as one of the leading causes of excessive weight. Age was also a strong predictor; an increase of 30% in the prevalence of being overweight/obese was noted with each decade. Of note, respondents in this study showed the highest knowledge score pertaining to diet and diseases (80.7%) [14]. This finding is very similar to our data, showing that diet alone is the most popular method among those who have ever attempted to lose weight (58% of the total sample and 84% of obese individuals).

Another major problem lies in the fact that even if generally perceived as a disease, obesity is not commonly treated. In our study sample, only 56% of those living with obesity had first consulted a medical specialist, with 26% consulting a fitness or gym coach. This tendency to rely on information gathered in the gym environment seems typical for our population. In the aforementioned study from the Lazio region, Italy, 57.7% of participants had obtained specific information from healthcare providers, followed by social media (19.1%) [22]. In the ACTION-FRANCE study, weight-related discussions with healthcare professionals were reported as “surprisingly infrequent” [28]. One of the main factors responsible for this finding was the fact that obese individuals perceived weight management as their personal responsibility or felt uncomfortable discussing it [28]. The situation in our Bulgarian participants was quite similar, with preferred medical specialists with whom to discuss their own weight being endocrinologists and nutritionists. An additional unexpected finding of our study is highlighted in the inefficiency of family support for people living with obesity. Only 16% of participants believed that they could gain support from their closest relatives when losing weight.

The next logical question concerning obesity covers treatment strategies and patient preferences. Among our respondents, the preferred methods were diet and physical activity (almost 80%), followed by the use of OTC drugs and supplements (77%), cosmetic procedures (65%), fitness/gym-based weight loss programs (60%), and surgical interventions (53%). Less than half of participants (48% only) stated that they would try drugs approved for weight loss. A similar situation was reported in the ACTION-FRANCE study [28]. Relevant discussions were held mainly with GPs and primarily focused on physical activity and diet. Of note, psychotherapy was also likely to be prescribed (55%), with pharmaceutical options rarely mentioned (noted in only 19.5% of cases). These gaps typical of discussions with primary care specialists regarding weight reduction were highlighted in a study summarizing experiences from the US [29]. The targeted education of primary care physicians is one of the more efficient strategies for weight management, as they are generally the most frequently visited medical specialists; in comparison, many GPs simply fail to register their patients’ overweight state, as shown by the results of a recent Polish study [30]. A systematic review exploring GPs’ and nurses’ perspectives on shared weight management with their patients revealed that obesity was often not seen as a priority and that other lifestyle interventions, such as quitting smoking, were often perceived as being of greater importance [31]. With regard to GPs’ and nurses’ perspectives of obesity, their clinical skills and knowledge might be insufficient. Moreover, regarding this topic, the results of one study revealed an alarming lack of understanding by medical students regarding sports and exercise medicine [32].

Another interesting finding of our study is the fact that keeping the lost weight off was considered a much more substantial result than primary weight loss. This is an observation rarely mentioned in clinical studies. The above represents an important issue, as body weight variability has been definitively linked with increased cardiovascular risk [33]. In addition, the considered benefits of obesity drugs as listed by our respondents were safety (73%), efficacy (69%), and positive impact on CV risk factors (54%). The perceived barriers included fear of weight increase following treatment discontinuation (63%), high drug prices (48%), fear of needles/injections (43%), and need for long-term use (40%). Examining these data, one must keep in mind that the urgency to act as well as the aggressiveness of the preferred treatment modalities seem to be in direct correlation with the extent of BMI excess [34]. Finally, but importantly, it should be noted that there are a great number of psychological issues related to individual obesity perception and willingness to take action, factors beyond the scope of the present study [35].

### Strengths and Limitations

The present study has some limitations. First, the study included participants from urban areas only, with a moderate-to-high income level. Secondly, the specific questionnaire itself and the multiple-choice questions might have introduced some bias in the recorded answers and selected preferences. Thirdly, the psychological characteristics of our respondents remain unexplored.

However, it should be noted that this study also has many strengths. It is the first of its kind in Bulgaria, revealing the scope of the problem as perceived by affected individuals. Even if participants correctly identify themselves as overweight/obese, they are often satisfied with their overall health status and do not perceive any urgency in losing weight. However, those who wish to lose weight may often harbor unrealistic expectations that can lead to severe disappointment. The causes of excessive weight gain are also poorly understood. The long-term management of obesity is still an enigma for the affected individuals, and modern anti-obesity drugs remain largely unavailable. Healthcare professionals are perceived by obese people as not efficient enough in correctly addressing weight problems and offering realistic approaches and management strategies, coupled with understanding, empathy, and effective support.

## 5. Conclusions

There exists a substantial gap between scientific/clinical knowledge of obesity and patients’ perspectives, attitudes, and expectations. Filling this gap requires diligent effort both in increasing public awareness and correct knowledge and in implementing new health strategies for large-scale clinical programs to combat the epidemic of obesity. More detailed studies are still needed to better understand the incentives for both patients and healthcare professionals.

## Figures and Tables

**Figure 1 nutrients-17-00373-f001:**
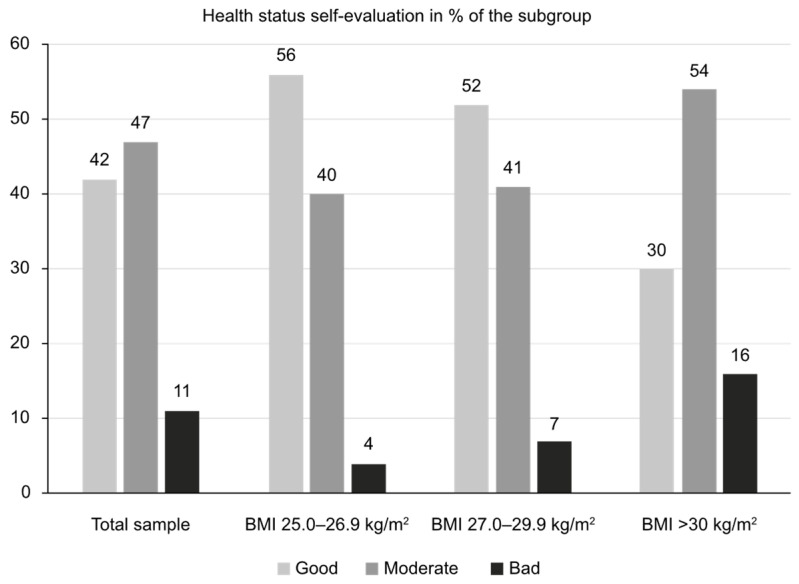
Overall health self-evaluation (“How would you evaluate your overall health condition?”).

**Figure 2 nutrients-17-00373-f002:**
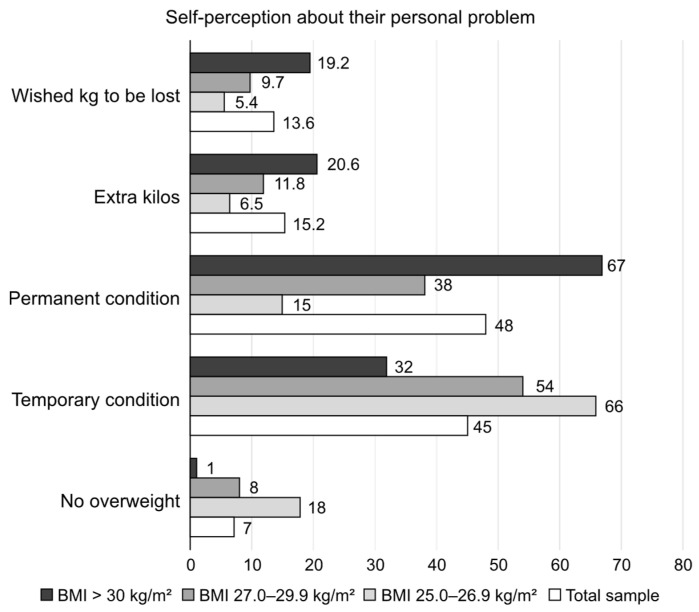
The self-perception of the respondents of being overweight/obese is shown as shares in the total sample and BMI subcategories (“Do you consider yourself having extra kg? How many? How many would you like to lose? Any extra weight?”).

**Figure 3 nutrients-17-00373-f003:**
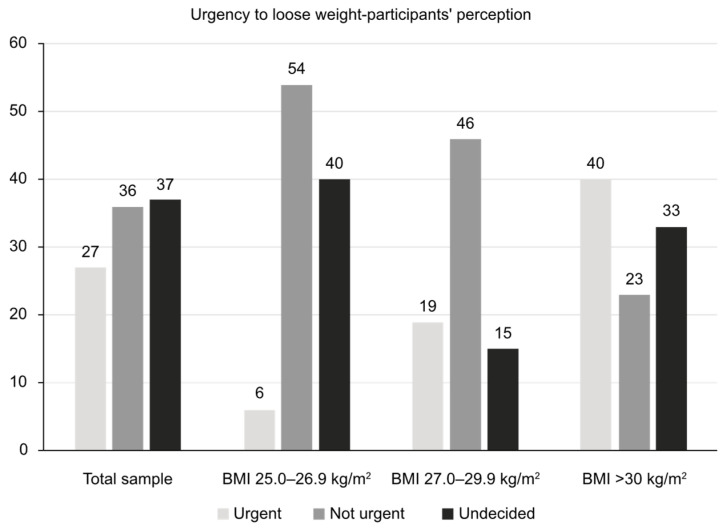
The need for urgent reduction in body weight as perceived by the respondents (shares are shown in % of the total).

**Figure 4 nutrients-17-00373-f004:**
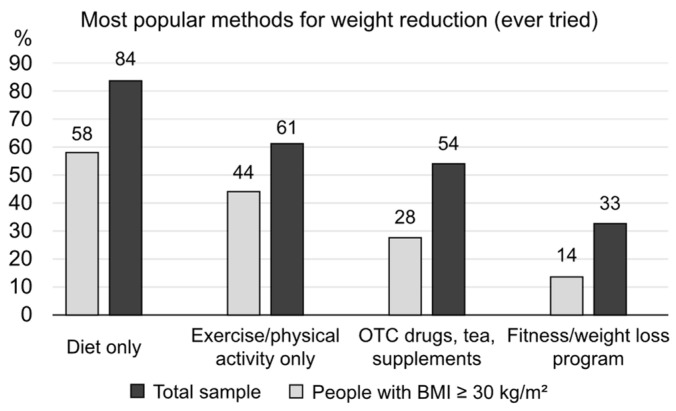
The most popular methods for weight reduction among those who had made previous attempts.

**Table 1 nutrients-17-00373-t001:** The sample structure and profile (in total and stratified according to BMI) are shown in %.

	Total Sample N = 704	BMI 25.0–26.9 kg/m^2^ N = 147	BMI 27.0–29.9 kg/m^2^ N = 197	BMI ≥ 30.0 kg/m^2^ N = 360
Sex				
Male/Female	44/56	35/65	46/54	47/53
Age				
25–34 y/o	21	25	23	19
35–44 y/o	29	33	26	28
45–54 y/o	28	29	30	27
55–64 y/o	22	13	21	26
Education				
Higher	50	49	51	51
Secondary	50	51	49	49
Family status				
Single	19	18	20	19
Married	58	55	58	59
In a partnership	18	27	22	22
Have children	43	45	45	44
No children	57	55	55	56
Employment				
Self-employed	12	9	10	13
Employee	85	90	83	84
Other	3	1	7	3

**Table 2 nutrients-17-00373-t002:** Different reasons for being overweight/obese in the BMI subcategories.

BMI 25.0–26.9 kg/m^2^	BMI 27.0–29.9 kg/m^2^	BMI ≥ 30.0 kg/m^2^
Stress or depression Overconsumption Hormonal changes/slow metabolism	High intake of carbs Irregular meals Lack of physical activity/exercise	Poor quality of food products in stores Sedentary lifestyle

**Table 3 nutrients-17-00373-t003:** Most trusted medical specialists for help with weight loss.

	Total Sample N = 704	BMI 25.0–26.9 kg/m^2^ N = 147	BMI 27.0–29.9 kg/m^2^ N = 197	BMI ≥ 30.0 kg/m^2^ N = 360
Nutritionist	25	23	18	29
Endocrinologist	17	14	11	21
Cardiologist	3	1	3	4
GP	19	20	24	15
Other	3	1	4	3

## Data Availability

The data presented in this study are available on request from the corresponding author due to privacy reasons.

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
