# Peer review of "The Perceptions of and Attitudes Toward Obesity in Bulgarian Adults with a BMI ≥ 25.0 kg/m2—An Exploratory Study"

_nutrients, 2025, doi:10.3390/nu17030373_

Round 1
Reviewer 1 Report
Comments and Suggestions for Authors
Manuscript ID: nutrients-3388488
The purpose of this study was to examine the views and attitudes toward obesity in Bulgarian adults who are overweight or obese.
Article Report
Comments and Suggestions for Authors
This article
I. Major comments:
1. Extend the introduction section and explain the consequences of obesity and which are the comorbidities associated with obesity. Add more references in the introduction section.
2. The discussion is a bit poor, rewrite it, comparing the study with previous similar studies, and add more references in the discussion section.
II. Minor comments:
1. Line 96 and 235, add the super index to square meter.
2. Line 139: replace comma per point
3. Figure 2: replace comma per point
4. Table 2: lack of physi-? Could you replace it with another word to better understand?
5. Lines 257,259, and 260 add the super index to square meters.

Quality of English
Reviewer 2 Report
Comments and Suggestions for Authors
Dear authors,
Boyanov et al. explored the perceptions and attitudes of adult Bulgarians towards overweight and obesity. This field clearly needs exploration, as it sheds light on aspects not frequently covered in the literature.
However, I have some comments and suggestions that could improve the manuscript.
Abstract
Please add the study design in Methods.
Please change the language from obese people to people with overweight or people with obesity throughout the manuscript.
Please change “consummation” for consumption.
Introduction
Please change some of the wording for a more scientific vocabulary throughout the manuscript. For instance, change “more and more” for “increasing”, or “a lot less” for “decreasing” or other alternatives.
Line 57: change “very few” for “little”
Materials and methods
I’m unsure regarding the point where the authors claimed that for this type of studies no ethical review is needed.
Please state the power and level of significance of the sample size calculation.
Results
No comments.
Discussion
In terms of the GP and nurses’ perspective on obesity, there might be some issues regarding their clinical skills and knowledge regarding this topic. For instance, this paper explores the lack of understanding of medical student regarding sports and exercise medicine: McGuire B, Mahfouz H, Lorenz H, Archer E. Sport and Exercise Medicine: a misunderstood specialty among medical students and foundation doctors. Int J Med Stud [Internet]. 2024 Nov. 19 [cited 2024 Dec. 16];. Available from: https://ijms.info/IJMS/article/view/2634
